# Effects of Different-Syllable Aggressive Calls on Food Intake and Gene Expression in *Vespertilio sinensis*

**DOI:** 10.3390/ani13020306

**Published:** 2023-01-15

**Authors:** Xin Li, Ruizhu Zhou, Lei Feng, Hui Wang, Jiang Feng, Hui Wu

**Affiliations:** 1College of Life Science, Jilin Agricultural University, Changchun 130118, China; 2Jilin Provincial Key Laboratory of Animal Resource Conservation and Utilization, Northeast Normal University, Changchun 130117, China; 3Key Laboratory of Vegetation Ecology of Education Ministry, Institute of Grassland Science, Northeast Normal University, Changchun 130117, China

**Keywords:** aggressive calls, social conflict, social stress, food intake, transcriptome

## Abstract

**Simple Summary:**

Most social animals have to face the social stress caused by territorial conflicts. To save costs, social animals use acoustic signals instead of physical fights to solve conflicts. Bats live in clusters and frequently produce aggressive calls of different syllables, but little is known about the effects of social stress represented by different types of aggressive calls on the physiology of bats. Here, we conducted playback experiments to investigate the effects of two types of aggressive calls representing different competitive intentions on food intake, body mass, hormone levels, and gene expression in Asian particolored bats (*Vespertilio sinensis*). Our results showed that different types of aggressive calls exerted different physiological effects on social animals. Interestingly, we found that more aggressive calls do not have a greater impact on bats.

**Abstract:**

Social animals enjoy colony benefits but are also exposed to social stress, which affects their physiology in many ways, including alterations to their energy intake, metabolism, and even gene expression. Aggressive calls are defined as calls emitted during aggressive conflicts between individuals of the same species over resources, such as territory, food, or mates. Aggressive calls produced by animals in different aggressive states indicate different levels of competitive intentions. However, whether aggressive calls produced in different aggressive states exert different physiological effects on animals has yet to be determined. Importantly, bats live in clusters and frequently produce aggressive calls of different syllables, thus providing an ideal model for investigating this question. Here, we conducted playback experiments to investigate the effects of two types of aggressive calls representing different competitive intentions on food intake, body mass, corticosterone (CORT) concentration, and gene expression in *Vespertilio sinensis*. We found that the playback of both aggressive calls resulted in a significant decrease in food intake and body mass, and bats in the tonal-syllable aggressive-calls (tonal calls) playback group exhibited a more significant decrease when compared to the noisy-syllable aggressive-calls (noisy calls) playback group. Surprisingly, the weight and food intake in the white-noise group decreased the most when compared to before playback. Transcriptome results showed that, when compared to the control and white-noise groups, differentially expressed genes (DEGs) involved in energy and metabolism were detected in the noisy-calls playback group, and DEGs involved in immunity and disease were detected in the tonal-calls playback group. These results suggested that the playback of the two types of aggressive calls differentially affected body mass, food intake, and gene expression in bats. Notably, bat responses to external-noise playback (synthetic white noise) were more pronounced than the playback of the two aggressive calls, suggesting that bats have somewhat adapted to internal aggressive calls. Comparative transcriptome analysis suggested that the playback of the two syllabic aggressive calls disrupted the immune system and increased the risk of disease in bats. This study provides new insight into how animals differ in response to different social stressors and anthropogenic noise.

## 1. Introduction

Many social animals, from small insects to large mammals, are found in nature [1,2]. Animals derive several benefits from social living, such as the reduced risk of being preyed upon and an increased success rate of predation [3,4]. However, the limited availability of resources, including water, mates, food, and space, leads to social conflict. The aggressive conflict between individuals of the same species is one of the most common sources of social stress [5,6,7]. Social stress exerts a range of effects on animals. For example, social stress activates the hypothalamic–pituitary–sympathetic adrenal (HPA) medulla system in animals [8]. HPA promotes the secretion of stress hormones, such as adrenocorticotropic hormone (ACTH) and corticosterone (CORT) [9,10,11,12]. These hormones work synergistically to relieve anxiety and aggressive emotions in animals under stress [10,13,14]. High CORT levels under stress scenarios stimulate appetite and lead to increased food intake, which in turn leads to weight gain [10,15,16]. However, changes in food intake and body mass caused by stress have not been consistently researched. Social stress has been shown to lead to decreased food intake and body mass in animals, such as weight loss in *Tupaia belangeri* after chronic social stress, which has been attributed to increased stress-induced metabolic activity [17], and to a lesser extent, reduced food intake [18]. Stress responses are fundamental survival mechanisms that allow animals to adapt to environmental challenges, but these responses may also harm their immune systems [19,20]. In addition, long-term social stress can significantly alter gene expression, while short-term stimulation results in fewer significant transcriptional effects [21].

Social conflict can be resolved in many ways. To reduce unnecessary energy consumption, social animals use acoustic signals instead of physical fights to resolve such conflicts [22,23]. Aggressive calls are produced during aggressive conflicts between individuals of the same species for resources, such as territory, food, or mates [24]. Calls in different aggressive states represent different degrees of aggression. The sounds emitted by invaders and those being invaded have different characters in their frequency and tonality. This phenomenon has been observed in many animal groups, including frogs, birds, and mammals [25,26,27]. The effects of stress on physiology, metabolism, and gene expression have attracted considerable attention and have become a research focus in the fields of biology and ecology [28]. However, it remains to be determined whether there are differences in the physiological effects of different intensities of social stress on the animal body as characterized by different syllables of aggressive calls. It is difficult to fully assess the physiological effects of different social stressors represented by different levels of aggression on animal physiology based on behavioral methods alone. The effects of aggressive calls on animal physiology require further comprehensive evaluation from the perspectives of metabolism, hormone levels, and gene expression.

Bats comprise the second-largest group of mammals, which are typically nocturnal, with poor vision and have a well-developed echolocation system. Accordingly, bats have long been used as model animals in acoustic research [29]. The ability to fly and emit echolocation has allowed many bats to evolve unique and complex acoustic communication systems [30]. In bat clusters, conflicts frequently occur between individuals for the central perch position; thus, they emit different-syllable aggressive calls [23]. To explore whether different intensities of social pressure represented by different-syllable aggressive calls exert different effects on animal physiology, we selected female Asian particolored bats (*Vespertilio sinensis*) as the research object. These female bats form large colonies during the breeding season and cluster together under an overpass, and frequently emit different-syllable aggressive calls to compete for habitat resources during the day. At night, they search for food and rarely emit aggressive calls. Previous research has shown that tonal-syllable aggressive calls (also called “tonal calls”) and noisy-syllable aggressive calls (also called “noisy calls”) reflect different competitive intentions in the social conflict resolution strategies of female *V. sinensis* [23]. One study showed that in a crowded roost of *V. sinensis*, bats pushed each other; residents frequently produced aggressive calls when they were pushed by intruders over their forearm or head for a better roosting site. When the residents emitted aggressive calls with more or a higher ratio of tonal syllables, the intruders continued to push them, but when they emitted aggressive calls with more or a higher ratio of noisy syllables, the intruders stopped pushing. This suggests that the number of noisy/tonal syllables in aggressive calls in *V*. *sinensis* is a signal of intent to attack [23]. Compared to tonal calls, noisy calls indicate higher aggressive intentions, and these acoustic signals may reduce physical contact and resolve conflicts in a timely manner to minimize energy consumption [23]. For bats, different-syllable aggressive calls convey different competitive intentions and may reflect different levels of social stress.

To investigate these questions, we performed a playback experiment in the lab, broadcasting different-syllable aggressive call files to two groups of bats using white-noise playback and silence (no playback) as the controls to compare changes in food intake, body mass, CORT concentrations, and gene expression of each group during the experiment. We hypothesized that the social pressure generated by the two aggressive calls would exert different effects on the bats and that noisy calls expressing higher aggressive intentions would have a greater effect than tonal calls. We predicted the following: (1) compared to the control and white-noise groups, the aggressive-call playback group would exhibit greater changes in body mass, food intake, and CORT hormone concentrations; (2) compared to the tonal-calls playback group, the noisy-calls playback group would exhibit greater changes in body mass, food intake, and CORT hormone concentrations; (3) compared to the control and white-noise groups, the number of differentially expressed genes (DEGs) in the noisy-calls playback group would be greater than the tonal-calls playback group. Genes or pathways associated with stress found in the noisy and tonal aggressive-call playback groups were compared to the control and white-noise groups.

## 2. Materials and Methods

### 2.1. Study Site and Species

In the summer of 2021, we captured 60 adult female *V*. *sinensis* using mist nets under an overpass located in Acheng, Heilongjiang Province, northeast China. A total of 36 bats were used for recording aggressive calls and 24 bats participated in the playback experiment. To avoid the reuse of the recording and playback bats, the 36 bats used for the recordings were not released until the 24 bats used in the playback experiments were caught. All bats were placed under a 12/12 h light/dark photoperiod with light starting at 8:00 am. Animals were allowed to freely feed on mealworms and water; a sufficient number of vitamins was added to the water. The temperature of the laboratory was 26 ± 2 °C and the relative humidity was 55 ± 5%.

### 2.2. Recording of Aggressive Calls and Editing of Playback Files

The 36 captured bats were randomly divided into three groups, then placed in three cages of the same size (0.5 m × 0.5 m × 0.5 m), which were marked as S1 (*n* = 12), S2 (*n* = 12), and S3 (*n* = 12). The right forearm of each bat was marked with metal rings (5.2 mm inner diameter, 5.5 mm height) (Porzana Ltd., Winchelsea, England) to identify individuals. The three cages were placed in different rooms to reduce habituation to the respective sounds. We used the “resident-intruder paradigm” method to record the aggressive calls [23,24]. A bat from cage S1 was placed in the experimental cage (0.5 m × 0.5 m × 0.5 m) and used as the “resident.” After the “resident” stabilized, we placed an “intruder” from cage S2 into the experimental cage. After placement of the intrusive bat in the experimental cage, we observed that the “intruder” would usually crawl to the “resident” and try to inhabit it. The “intruder” would push with its head or forearm to compete with the “resident” for a perching position close to the center. The disturbed bat often displayed aggressive behavior and emitted aggressive calls. Therefore, the calls recorded using the “resident-intruder paradigm” method were considered aggressive. To ensure the quality of the recordings, we placed sound-absorbing sponges on the walls of the recording studio. If two bats did not emit a sound within 10 min, the recording was terminated. In the recording experiment, the bats used as “residents” or “intruders” came from different cages. On the same day, a bat was used only once as a “resident” or “intruder”; the same method was used to record the aggressive calls of other bats. Aggressive calls were recorded using Avisoft UltrasoundGate 116H (Avisoft Bioacoustics, Berlin, Germany) with a condenser ultrasound microphone (CM16/CMPA; Avisoft Bioacoustics). The sampling frequency was 250 kHz with 16 bits. After completing the recording experiments, the 36 bats were released to inhabit the original roost. All bats were healthy and able to fly upon release.

We selected the recorded aggressive calls of 30 individuals and edited the playback files using the sonic analysis software, Avisoft-SASlabPro (Avisoft Bioacoustics). A syllable is the smallest independent unit surrounded by whitespace in a sentence [30]. A sentence refers to a sequence consisting of one or more syllables. A syllable with noisy components was used as the noisy syllable (noisy calls) and a syllable with only tonal syllable components was used as the tonal syllable (tonal calls) [24,31,32]. Each playback file consisted of 15 sentences with a strong signal-to-noise ratio (>20 dB) randomly selected from 15 individuals (one sentence per individual) with a length of 2.5 s. All playback files had a syllable interval of 0.009 s, a sentence interval of 0.094 s, and an interval of 17.5 s [23]. There were four playback files consisting of tonal and noisy aggressive calls from 30 individuals, and the calls in the four playback files were reused (Figure 1); the individuals in the tonal-calls playback files corresponded to the noisy-calls playback files one by one. Tonal- and noisy-calls playback files 80 s in duration were edited. We created an artificially synthesized playback file from a set of white noise. The white-noise parameters were as similar as possible to *V. sinensis* aggressive call syllables in a natural state. The lowest and highest frequencies were set to 7.6 and 61.8 kHz, respectively. The energy ranged from 7.6–20 kHz and the sound gradually decreased from 20 to 61.8 kHz. Finally, all playback files were normalized so that the amplitude of the weakest sentence in the file was >30 dB [23].

### 2.3. Playback Experiment

To reduce capture pressure, playback experiments began after the bats acclimated to the lab for two weeks. The 24 bats were divided into four groups of similar size: the noisy-calls playback group (group A), tonal-calls playback group (group B), white-noise playback group (group C), and control group (group D). Each bat was placed individually in a cage (0.3 m × 0.3 m × 0.3 m) and bats from the four groups were placed in four playback labs with similar environments. All bats did not hear playback for 11 d, then speakers (Ultrasonic Dynamic Speaker, Vifa) (Ultra Sound Gate Player 116; Avisoft Bioacoustics) were used to play noisy and tonal calls or white noise to bats in groups A, B, and C; group D did not undergo any treatment. The speakers were placed ~1 m away from the center of the cage where the bats were located. The sound pressure level was 70 dB from the speaker to the center of the cage. During the playback experiment, the playback was conducted from 8:30 to 18:30 every day. Noisy calls, tonal calls, and white noise were broadcasted to groups A, B, and C, respectively. Bats were provided with mealworms and water at 19:00 every day. The food intake and body mass of each bat in the four groups were measured using an electronic balance (LC-50 ProScale) accurate to 0.01 g. The daily food intake was calculated as the difference between the weight of mealworms offered and uneaten mealworms. The food intake was measured every day and the weight was measured every two days. Feces were collected from each bat to measure the CORT hormone levels before (11 d), during (12 d and 18 d), and after (25 d) playback. Fecal samples were kept on dry ice. CORT hormones were uniformly measured after the last collection. The CORT hormone levels were measured using a kit with a detection range of 2–80 μg/L. A double antibody sandwich assay was applied to determine the CORT levels in the specimen. CORT antibodies were added to the wells, and then combined with Horseradish Peroxidase (HRP)-labeled CORT antibodies to form antibody–antigen–enzyme-labeled antibody complexes, which were washed thoroughly and developed with trimethylolpropane (TMB). TMB was converted to blue using HRP enzymes and to yellow by adding acid. The shade of color positively correlated with the CORT level in the sample. The absorbance was measured at an optical density (OD) of 450 nm using an enzyme marker. The CORT concentration in the samples was determined by a standard curve. Then, we took the concentration of the standard as the horizontal coordinate and the OD value as the vertical coordinate, drew a standard curve on the coordinate paper and found the corresponding concentration from the standard curve according to the OD value of the sample. After multiplying by the dilution multiple or using the concentration of the standard and OD value to calculate the linear regression equation of the standard curve, we substituted the OD value of the sample in the equation, calculated the concentration of the sample, then multiplied by the dilution multiple, which was the actual concentration of the sample. We compared the daily food intake, body mass, and fecal CORT concentrations between groups A, B, C, and D using the lmer function in the lme4 R package [33]. The model included “group” (four categories: groups A, B, C, and D) and “playback” (two categories: on and off) as fixed factors. The interaction was between “group” and “playback” using “day” as a covariate and “bat ID” as a random factor. The data were visualized using the ggplot2 package, which can handle a large number of data types [34].

Kidney tissues were collected from three randomized individual bats in each group immediately at the end of the playback. Each tissue was individually placed in 2.0 mL RNase-Free (Biotech, Shanghai, China) lyophilization tubes, snap frozen in liquid nitrogen, and later refrigerated in an ultra-low temperature refrigerator at –80 °C until total RNA was extracted. See the Appendix A for details on RNA extraction and sequencing.

### 2.4. Differential Gene Expression Analysis

Gene expression levels of the samples were estimated using RSEM (RNA-Seq by Expectation Maximization, rsem-1.2.0) [35]. Clean data were matched with the assembled transcriptome. The expression level of each transcript was calculated using the transcripts per million reads (TPM) method [36]. Differential expression analysis of the kidneys of each group was performed using the DEG-seq2 package with biological replicates [37]. Genes with an adjusted *p*-value of ≤0.05 and |log2fold change| > 2 were set as the threshold for significantly differential expression.

### 2.5. GO/KEGG Enrichment Analysis for DEGs

In addition, functional-enrichment analyses, including GO and KEGG, were performed to identify which differentially expressed genes (DEGs) were significantly enriched in GO terms and metabolic pathways at *p*-values of ≤0.01 [35]. GO functional enrichment and KEGG pathway analyses were carried out by Goatools (https://github.com/tanghaibao/Goatools) (accessed on 21 June 2022) and KOBAS (http://kobas.cbi.pku.edu.cn/home.do) (accessed on 10 July 2022) [38].

## 3. Results

### 3.1. Effects of Aggressive Calls on Food Intake, Body Mass, and CORT Levels

Groups A, B, and C fed less after playback than before with average decreases of 2.7%, 13.4%, and 15.7%, respectively. Group D feeding increased on average by 13%. Similarly, the body mass in groups A, B, and C were all lower after playback than before, with average body mass decreases of 4.7%, 5.2%, and 13.9%, respectively. The average body mass of group D increased by 5.6%. The mixed-effects model showed that the interaction between “group” and “playback” had a significant effect on daily food intake and body mass, indicating that the two aggressive calls significantly affected food intake and body mass. During playback, the CORT levels in the two aggressive-call groups were higher than before playback, but the mixed-effects model showed that none of the variables had a statistically significant effect on the CORT hormone levels in each group (Table 1).

The K–W test results showed that after playback, the food intake in group A was significantly lower than that in group D. Before playback, there was no significant difference in food intake between groups C and B, while after playback, the food intake in group B was significantly lower than that in group C (Figure 2). There was no significant difference detected between the body mass in the four groups before playback, but significant differences were detected after playback (Figure 3). We did not detect significant changes in the CORT levels after playback (Figure 4). These results suggested that the playback of different-syllable aggressive calls significantly affected body mass and food intake in bats, and that group B exhibited a greater decrease in body mass and food intake than group A, in which, white noise had the greatest effect on body mass and food intake.

### 3.2. Effects of Aggressive Calls on Gene Expression

The transcriptome sequencing results and details of the mean raw and mean clean reads in the four groups are shown in Appendix A. A total of 195,827 unigenes were obtained by assembling clean reads using Trinity software. Unigene sequences were used as the reference genome in the later analysis of the kidney tissue. The unigenes had an average length of 1151 bp, with N50 at 1979 bp and N90 at 3491 bp, and most of the lengths were concentrated at 200 to 500 bp. Functional annotation of the already spliced and assembled genes was performed to obtain the functional information of the genes. The unigenes obtained were annotated with six major databases. The results show that 40,527 unigenes were successfully matched in the NR database, and a variable number of unigenes was annotated to the other databases. In summary, a total of 42,577 unigenes were annotated by at least one database (Appendix A).

Based on the screening criteria, a total of 519 DEGs were identified in the six comparisons: group A vs. B, group A vs. C, group A vs. D, group B vs. C, group B vs. D, and group C vs. D (Table 2). The highest number of DEGs was detected in the group A vs. C comparison with 192 genes, followed by the group B vs. C comparison with 159 genes. In the group A vs. B comparison, the number of DEGs was the lowest, with 45 genes. This result suggested that the playback of the two aggressive calls differed in their effects and noisy calls exerted a greater effect than tonal calls. The hierarchical clustering heatmap of the 519 DEGs showed that there were significant differences in gene expression of the kidney tissues obtained from different groups (Figure 5).

### 3.3. GO and KEGG Enrichment Analyses of Differential Genes

To further understand the function of the DEGs, the DEGs of each comparison were enriched in the GO and KEGG databases. In the group A vs. D comparison, the GO enrichment analysis revealed a total of 139 terms that were significantly enriched with 121, 2, and 16 GO terms in the biological process (BP), cellular component (CC), and molecular function (MF) categories, respectively. In the BP category, DEGs were significantly enriched in the regulation of the multicellular organismal process, chemical homeostasis, and regulation of the developmental process. In the CC category, DEGs were significantly enriched in extracellular space, extracellular region, and MICOS complex. In the MF category, DEGs were significantly enriched in neurohypophyseal hormone activity, glyceraldehyde oxidoreductase activity, and oxytocin receptor binding. Per the KEGG results, DEGs were enriched in the folate biosynthesis and glycerolipid metabolism pathways. In the group A vs. C comparison, the GO enrichment analysis revealed a total of 141 terms that were significantly enriched, with 96, 23, and 22 GO terms in the BP, CC, and MF categories, respectively. In the BP category, DEGs were significantly enriched in the lipoprotein metabolic process, nicotinamide nucleotide metabolic process, and pyridine nucleotide metabolic process. In the CC category, DEGs were significantly enriched in the extracellular region, plasma membrane-bounded cell projection cytoplasm, and neuron projection cytoplasm. In the MF category, DEGs were significantly enriched in monovalent cation transmembrane transporter activity, transferase activity, transferring sulfur-containing groups, and lipid binding. Per the KEGG results, DEGs were significantly enriched in eight pathways, which were mainly related to digestion and metabolism. In the group B vs. D comparison, 245 terms were significantly enriched with 195, 21, and 29 GO terms in the BP, CC, and MF categories, respectively. In the BP category, DEGs were significantly enriched in biological processes, biological regulation, and regulation of biological processes. In the CC category, DEGs were significantly enriched in the endoplasmic reticulum, plasma membrane-bounded cell projection, and cell projection. In the MF category, DEGs were significantly enriched in molecular function regulators, lipid binding, and identical protein binding. Per the KEGG results, DEGs were significantly enriched in 17 pathways, including six organism system pathways, six human disease pathways, three environmental information processing pathways, one biological process pathway, and one metabolic pathway, of which, the number of immune system-related pathways was the highest with seven pathways, followed by five disease-related pathways; the other pathways were related to signal molecule interactions, transport, and catabolism, and metabolic synthesis of other secondary organisms. In the group B vs. C comparison, the GO enrichment analysis revealed that a total of 164 terms were significantly enriched with 116, 15, and 33 GO terms in the BP, CC, and MF categories, respectively. In the BP category, DEGs were significantly enriched in positive regulation of cell cycle phase transition, regulation of insulin secretion involved in the cellular response to a glucose stimulus, and regulation of peptide hormone secretion. In the CC category, DEGs were significantly enriched in the spindle microtubule, basolateral plasma membrane, and DNA topoisomerase type II (double strand cut, ATP-hydrolyzing) complex. In the MF category, DEGs were significantly enriched in protein tyrosine phosphatase activity, protein tyrosine/serine/threonine phosphatase activity, and phosphoprotein phosphatase activity. Per the KEGG results, DEGs were significantly enriched in pathways related to the endocrine and digestive systems and metabolism. In the group A vs. B comparison, 48 terms were significantly enriched with 34, 6, and 8 terms in the BP, CC, and MF categories. In the BP category, DEGs were significantly enriched in the regulation of the multi-organism process, negative regulation of glucagon secretion, and regulation of the G protein-coupled receptor signaling pathway. In the CC category, DEGs were significantly enriched in the extracellular region, endosome-to-plasma membrane transport vesicle, and epidermal lamellar body. In the MF category, DEGs were significantly enriched in prostaglandin-D synthase activity, apelin receptor binding, and prostaglandin E receptor activity. Per the KEGG results, DEGs were significantly enriched in three pathways, including ferroptosis, mineral absorption, and arachidonic acid metabolism. In the group C vs. D comparison, 59 terms were significantly enriched with 39, 4, and 16 terms in the BP, CC, and MF categories. In the BP category, DEGs were significantly enriched in protein transport within the extracellular region, DNA cytosine deamination, and intermediate-density lipoprotein particle clearance. In the CC category, DEGs were significantly enriched in plasma membrane proton-transporting V-type ATPase complex, low-density lipoprotein particles, and intermediate-density lipoprotein particles. In the CC category, DEGs were significantly enriched in vitamin E binding, STAT family protein binding, and nitrate transmembrane transporter activity. According to the KEGG results, DEGs were significantly enriched in two pathways, including complement and coagulation cascades and bile secretion. Physiological processes related to digestion and metabolisms were altered in group A, while biological regulation and energy-related changes were detected in group B. The GO terms were significantly enriched in the MF, CC, and BP categories for each comparison group (Figure 6). The KEGG pathways were significantly enriched by the DEGs (Table 3). These results indicated that the playback of noisy and tonal calls exerted different effects on gene expression in bats.

## 4. Discussion

Social stress exerts a range of effects on animals, but the physiological effects of aggressive calls remain unknown. In this study, for the first time, the effects of two different-syllable aggressive calls on body mass, food intake, CORT concentration, and gene expression in *V. sinensis* were investigated. We found that, when compared to the control group, bats exposed to both aggressive calls ate less and weighed less. Surprisingly, the white-noise playback group exhibited the greatest change in body mass and food intake. Compared to the tonal-calls playback group, the noisy-calls playback group did not show greater changes in body mass, food intake, or CORT hormone concentration. In contrast, bats in the tonal-calls playback group showed greater changes. The genes and pathways associated with stress were detected in both the noisy- and tonal-calls playback groups when compared to the control and white-noise groups. These results did not conform to our predictions, indicating that our hypothesis was not supported.

Stress responses lead to appetite suppression and decreased food intake [39]. Increased conflict within populations increases stress levels and these elevated stress levels deplete energy and potentially impair immune functioning [40]. Intermittent tethered activation of sympathetic stress has a positive effect on maintaining hepatic lipid metabolic homeostasis, especially under high-fat diet conditions, which reduce energy intake [41]. Data on experimental rats were similar to those on humans who cope with stress by eating less [42,43]. In our study, compared to the control group, the two different types of aggressive-call playbacks resulted in decreased body mass and food intake. When compared to the tonal-calls playback group, the noisy-calls playback group did not exhibit changes in body mass, food intake, or CORT concentration, which was more than expected by chance. One explanation may be that, when compared to the tonal-calls playback group, the noisy-calls playback group compensated for the decrease. We observed an upward trend in daily food intake and body mass in the control group. These results suggest that the decrease in food intake and body mass in the two aggressive-call groups were mainly caused by playback stimulus, while the increase in body mass and food intake in the control group may be related to the feeding season or unrestricted food resources. No significant differences were detected in the CORT concentration among the four groups before or after playback; thus, the pressure caused by playback may not be enough to lead to statistical differences in hormone levels. However, the CORT concentration in the two aggressive-call playback groups was higher than before playback, indicating that the playback of aggressive calls exerted some pressure on the bats. The CORT concentration results corroborated the body mass and food intake results in the four groups.

The GO and KEGG enrichment analyses showed that the playback of noisy calls led to a stress response in bats and exerted effects on life processes, such as metabolism and signaling pathways, which in turn affected food intake, body mass, and hormone secretion. In contrast, the playback of tonal calls also led to stress responses in bats and exerted effects on life activity processes, such as immunity, disease, and metabolism (described below); thus, several important KEGG pathways and key DEGs may be involved in mediating the effects of different-syllable aggressive calls in bats.

In the group A vs. D comparison, DEGs were significantly enriched in the folate biosynthesis and glycerolipid metabolism pathways, suggesting that bat metabolism is affected by noisy-calls playback and that *AKR1B* serves important functions in regulating glycolipid metabolism and adipose tissue homeostasis [44]. Among them, lipoprotein lipase (*LPL*) genes were highly expressed in the noisy-calls playback group and were significantly enriched in metabolism-related entries and pathways. LPL is a key enzyme in lipid storage and the metabolism of many tissues [45], and has a facilitative effect on lipid uptake in mouse kidneys [46]. In the group A vs. D comparison, the high expression of *AKR1B* and *LPL* suggested that lipid metabolism was elevated in bats stimulated by noisy calls, which may be a factor in the lower body mass of this group. In the group A vs. C comparison, DEGs were significantly enriched in the cholesterol metabolism and fat digestion and absorption pathways. APOA1 is the main protein found in cholesterol particles and is associated with reduced fat and carbohydrate intake [47]. *lIPC* is an important gene that determines cholesterol concentration and is associated with energy metabolism [48]. *APOC3* and *APOA1*, which were lowly expressed in the noisy-calls playback group, may be responsible for the reduced body mass and food intake in bats as well. Collectively, the DEGs involved in energy and metabolism implied that the playback of noisy calls altered body mass and food intake and is thus associated with reduced body mass and food intake in bats.

In the group B vs. D comparison, the DEGs were significantly enriched in immune-, disease-, and metabolic-related pathways, of which, the chemokine signaling pathway plays an important role in triggering the chemotaxis of immune cells [49]. In this study, several upregulated genes were significantly enriched in the chemokine signaling pathway and these genes play important roles in the regulation of immune responses. The cytokine–cytokine receptor interaction (CCRI) plays an important regulatory role in immune and inflammatory responses [50,51]. Most of the DEGs enriched in the CCRI pathway were associated with immune regulation, inflammation, and host defense processes. In the group B vs. D comparison, the expression levels of these genes were significantly upregulated, including *CCL5*, *CD4*, *CD183*, *CCL8*, *CX3CR1*, and *CD261_2*. Therefore, we hypothesized that tonal calls would prompt the organisms to intensify their immune and inflammatory responses. Cytokine interactions lead to anorexia in acute and chronic diseases [50,51,52], which may cause reduced body mass and food intake in bats. We also found that the disease-related pathways associated with pathogenic *Escherichia coli* infection, leishmaniasis, chagas disease, influenza A, and staphylococcus aureus infection were significantly enriched by the DEGs, indicating that the immune homeostasis in group B was affected, thus increasing the risk of disease in the organisms. In the group C vs. D comparison, complement and coagulation cascades were significantly enriched by upregulated genes, including *CD21*. This is an important pathway related to immunity [53], as *CD21* plays an important role in immunity [54]. Bile secretion was significantly enriched by upregulated genes, including *OSTB*, which promotes the secretion of bile and affects digestion and absorption [55]. These results suggested that white-noise playback affected the digestive and immune systems in bats when compared to silence.

## 5. Conclusions

Our findings have important ecological implications for understanding the physiological responses of animals to aggressive calls. Particularly, it was clear that different-syllable aggressive calls exerted different physiological effects on social animals. The results indicated that vocal and highly clustered animals were somewhat adaptive to their own internal social stress signals and more sensitive to external noise stimuli. The playback of two types of aggressive calls resulted in a significant decrease in food intake and body mass, and these decreases in the tonal-calls playback group were more pronounced than in the noisy-calls playback group. When compared to the control and white-noise groups, multiple DEGs involved in energy and metabolism were detected in the noisy-calls playback group, while DEGs involved in immunity and disease were detected in the tonal-calls playback group. Future studies should investigate the neuropeptides involved in the regulation of energy homeostasis to determine how the nervous system of the animal brain responds to aggressive calls and thus affects the body mass and food intake of animals.

## Figures and Tables

**Figure 1 animals-13-00306-f001:**
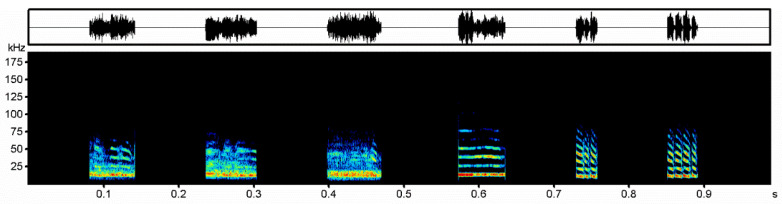
Aggressive calls of *Vespertilio sinensis*; the three on the left are noisy syllable (noisy calls), and the three on the right are tonal syllable (tonal calls).

**Figure 2 animals-13-00306-f002:**
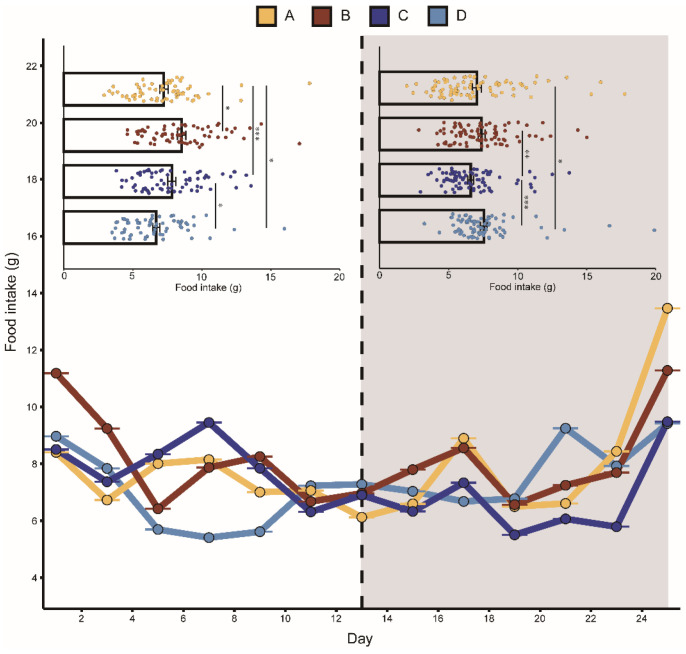
Change in food intake of bats in different groups. The playback period is represented by the dark grey box. A represents the noisy-calls playback group, B represents the tonal-calls playback group, C represents the white-noise playback group, and D represents the control group. The bar chart shows the results of the Kruskal–Wallis test, and the data are presented as mean ± standard error of the mean. * means *p* < 0.05, ** means *p* < 0.01, and *** means *p* < 0.001.

**Figure 3 animals-13-00306-f003:**
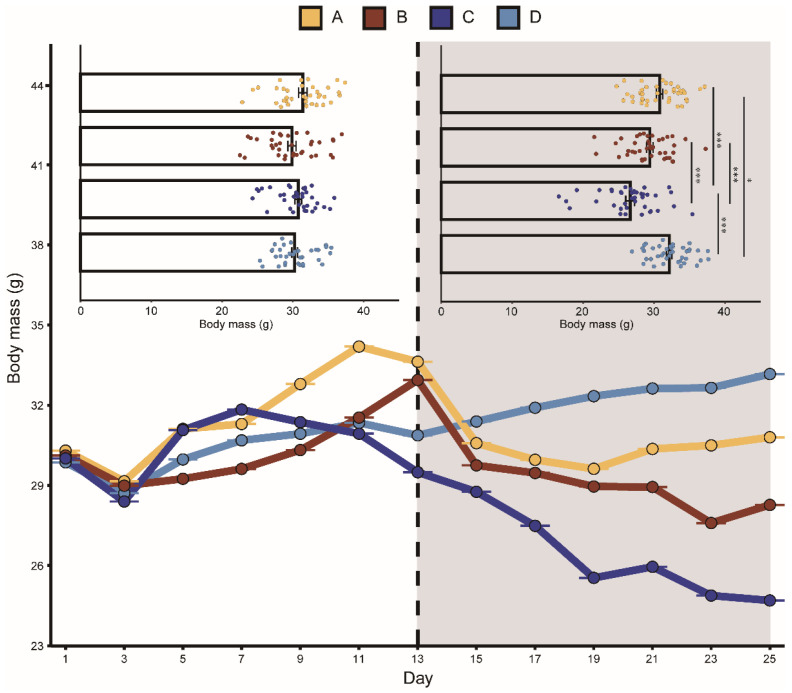
Change in body mass of bats in different groups. The playback period is represented by the dark grey box. A represents the noisy-calls playback group, B represents the tonal-calls playback group, C represents the white-noise playback group, and D represents the control group. The bar chart shows the results of the Kruskal–Wallis test, and the data are presented as mean ± standard error of the mean. * means *p* <0.05, and *** means *p* < 0.001.

**Figure 4 animals-13-00306-f004:**
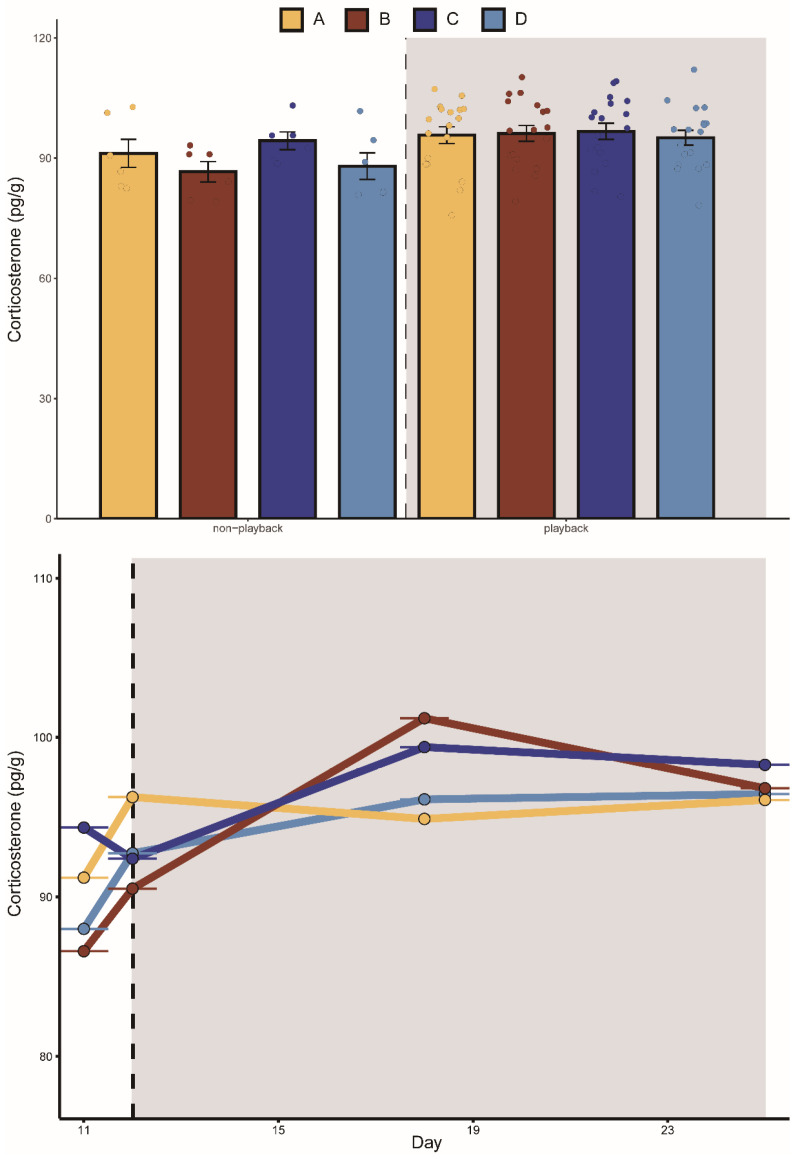
Change in corticosterone concentration of bats in different groups. The playback period is represented by the dark grey box. A represents the noisy-calls playback group, B represents the tonal-calls playback group, C represents the white-noise playback group, and D represents the control group. The bar chart shows the results of the Kruskal–Wallis test, and the data are presented as mean ± standard error of the mean.

**Figure 5 animals-13-00306-f005:**
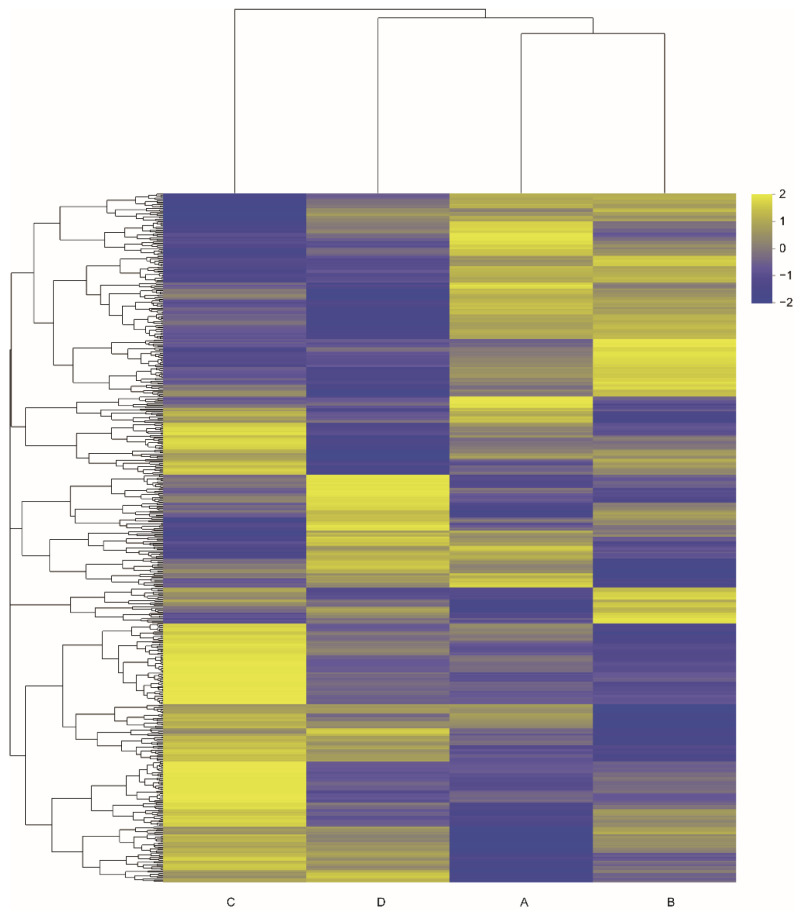
Heatmaps based on differentially expressed genes from six pairwise comparisons of four groups. A: the noisy-calls playback group, B: the tonal-calls playback group, C: the white-noise playback group, and D: the control group.

**Figure 6 animals-13-00306-f006:**
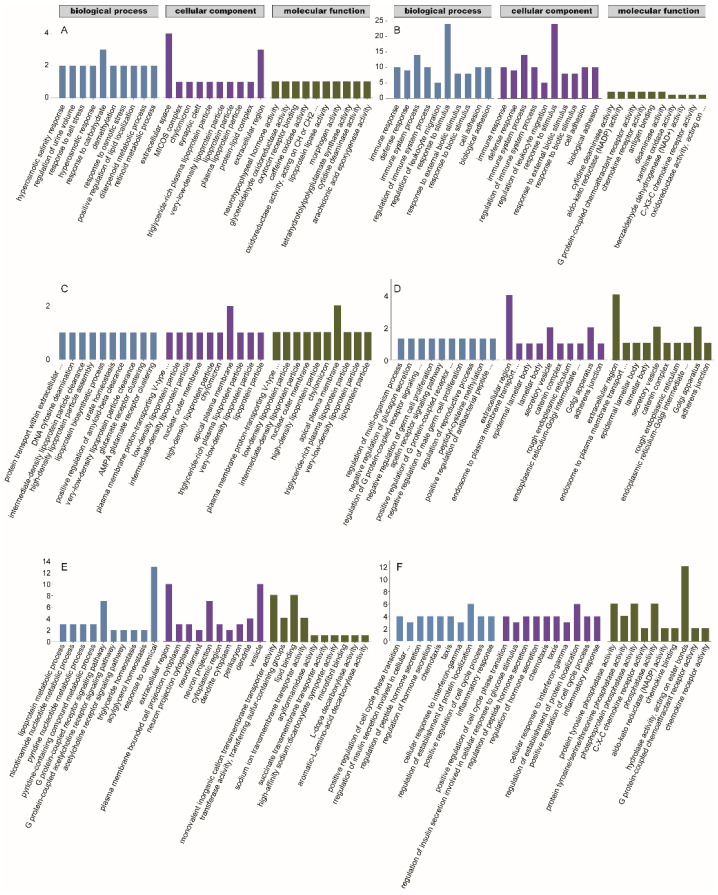
Gene ontology (GO) classifications of differentially expressed genes (DEGs) of different comparative groups. (**A**): the noisy-calls playback group vs. the control group; (**B**): the tonal-calls playback group vs. the control group; (**C**): the white-noise playback group vs. the control group; (**D**): the noisy-calls playback group vs. the tonal-calls playback group; (**E**): the noisy-calls playback group vs. the white-noise playback group; (**F**): the tonal-calls playback group vs. the white-noise playback group. X-axis: three major functional categories of GO terms, namely biological process, cellular component, and molecular function, Y-axis: the numbers of DEGs; the scale range of the Y-axis varies considerably from A to F.

**Table 1 animals-13-00306-t001:** Results of mixed-effect models on daily food intake, body mass, and fecal CORT hormone levels of bats in four groups.

	Food Intake			Body Mass			Corticosterone		
	Estimate (β ± SE)	*t*	*p*	Estimate (β ± SE)	*t*	*p*	Estimate (β ± SE)	*t*	*p*
Random effects									
Bat individual identity	0.91			2.38			0.00		
Residual	2.21			2.31			8.10		
Fixed effects									
Intercept	6.45 ± 0.48	13.32	<0.001	30.33 ± 1.06	28.62	<0.001	84.80 ± 3.85	22.00	<0.001
Day	0.04 ± 0.02	1.65	0.10	−0.01 ± 0.03	−0.42	0.67	0.29 ± 0.18	1.61	0.11
Group A	0.55 ± 0.65	0.85	0.40	1.22 ± 1.47	0.83	0.41	3.21 ± 4.68	0.69	0.49
Group B	1.84 ± 0.65	2.81	0.08	−0.28 ± 1.47	−0.19	0.85	−1.4 ± 4.68	−0.30	0.76
Group C	1.13 ± 0.65	1.74	0.09	0.35 ± 1.47	0.24	0.81	6.35 ± 4.68	1.36	0.18
Playback	0.35 ± 0.47	0.73	0.46	2.07 ± 0.66	3.11	0.002	4.98 ± 4.04	1.23	0.22
Group A: Playback	−1.05 ± 0.51	−2.04	0.04	−2.58 ± 0.71	−3.64	<0.001	−2.57 ± 5.40	−0.47	0.63
Group B: Playback	−1.20 ± 0.51	−3.88	<0.001	−2.44 ± 0.71	−3.43	<0.001	2.48 ± 5.40	0.46	0.65
Group C: Playback	−2.08 ± 0.51	−4.04	<0.001	−5.80 ± 0.71	−8.16	<0.001	−4.8 ± 5.40	−0.88	0.38

Group A: noisy-calls playback group, Group B: tonal-calls playback group, Group C: white-noise playback group.

**Table 2 animals-13-00306-t002:** The number of DEGs in the six pairwise comparisons.

Comparative Groups	Up	Down	Total
A vs. D	28	43	71
B vs. D	69	38	107
C vs. D	51	19	70
A vs. B	18	27	45
A vs. C	58	134	192
B vs. C	55	104	159

**Table 3 animals-13-00306-t003:** Kyoto Encyclopedia of Genes and Genomes pathways of differentially expressed genes between bats in different comparative groups.

Pathway Described	ID	*p*-Value	Number of DEGs
A vs. D			
Folate biosynthesis	ko00790	0.001094	2
Glycerolipid metabolism	ko00561	0.006939	2
C vs. D			
Complement and coagulation cascades	ko04610	0.006103	2
Bile secretion	ko04976	0.007234	2
A vs. B			
Arachidonic acid metabolism	ko00590	0.001225	2
Ferroptosis	ko04216	0.001382	2
Mineral absorption	ko04978	0.002842	2
B vs. C			
Bile secretion	ko04976	0.000972	4
Progesterone-mediated oocyte maturation	ko04914	0.001312	4
Oocyte meiosis	ko04114	0.002818	4
Mineral absorption	ko04978	0.007589	3
Platinum drug resistance	ko01524	0.009405	3
Phenylalanine metabolism	ko00360	0.009523	2
A vs. C			
Cocaine addiction	ko05030	0.000228	4
Mineral absorption	ko04978	0.000632	4
Steroid hormone biosynthesis	ko00140	0.001166	3
Cholesterol metabolism	ko04979	0.001601	3
Alcoholism	ko05034	0.005462	4
Amphetamine addiction	ko05031	0.008076	3
Phenylalanine metabolism	ko00360	0.009040	2
Fat digestion and absorption	ko04975	0.009533	2
B vs. D			
Viral protein interaction with cytokine and cytokine receptor	ko04061	0.000005	5
Chemokine signaling pathway	ko04062	0.000051	6
Leishmaniasis	ko05140	0.000065	5
Cytokine–cytokine receptor interaction	ko04060	0.000085	6
Cell adhesion molecules	ko04514	0.000421	5
Pathogenic Escherichia coli infection	ko05130	0.000515	6
Th1 and Th2 cell differentiation	ko04658	0.000599	4
NOD-like receptor signaling pathway	ko04621	0.000619	5
Hematopoietic cell lineage	ko04640	0.000671	4
Leukocyte transendothelial migration	ko04670	0.000876	4
Chagas disease	ko05142	0.000876	4
Rheumatoid arthritis	ko05323	0.001321	4
Influenza A	ko05164	0.003221	4
Staphylococcus aureus infection	ko05150	0.006513	3
Th17 cell differentiation	ko04659	0.007996	3
Caffeine metabolism	ko00232	0.008818	1
Phagosome	ko04145	0.009057	4

A vs. D: the noisy-calls playback group vs. the control group; B vs. D: the tonal-calls playback group vs. the control group; C vs. D: the white-noise playback group vs. the control group; A vs. B: the noisy-calls playback group vs. the tonal-calls playback group; A vs. C: the noisy-calls playback group vs. the white-noise playback group; B vs. C: the tonal-calls playback group vs. the white-noise playback group.

## Data Availability

All data generated or analyzed during this study are included in the published article and its additional files.

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
