# Peer review of "Effects of Different-Syllable Aggressive Calls on Food Intake and Gene Expression in Vespertilio sinensis"

_animals, 2023, doi:10.3390/ani13020306_

Round 1
Reviewer 1 Report
This is a potentially interesting paper using playback of two types of aggressive calls to captive bats to show the different calls induced changes in metabolism, food intake, body weight and gene expression. The paper is generally well written and interesting. However, there are several points that need further exploration before the paper can be full evaluated.
I was confused by the initial use of “syllables” since this can have both a quantitative (how many?) and qualitative (different acoustic structure) meaning. I would drop the term “syllable” and describe the two calls as “noisy calls” and “tonal calls” throughout. This is much clearer to readers.
The authors do not help us distinguish between the contexts in which the two types of calls are given. More behavioral detail about these contexts would help interpret the results. Is one type used in mild aggression and the other in contexts leading to physical aggression? This additional information is important to include.
In the methods were animals housed individually or in groups? If individually, the contexts for the different calls may not be natural. If group housed, the cage dimensions of 0.5m3 seem quite small for group housing.
The protocol for presenting the playbacks is not described. Were these continuous for 24h a day as the figure implies or were they played episodically? If the latter, was the playback pattern representative of the natural use of these calls? This needs to be clarified. If the playback supersaturated the normal rate of calling one would expect increased levels of stress, but these might not be natural but rather artifacts of the methods used.
I’m not competent to evaluate the section on gene expression but hope that another reviewer can address this better. Many comparisons are made between different playback conditions but were corrections made for multiple statistical tests? Functional labels are given for the various genes, presumably based on function in humans, but do we know if these same genes have a similar function in the bats?
In the discussion the authors are surprised by the greater response to white noise compared to call playbacks, but white noise is presumably novel for the bats, whereas the calls are familiar and have a normal context, so this might lead to greater stress response to white noise.
The authors make reference to Morton’s motivational-structural rules in the Introduction. Do these rules lead to different predictions about the two types of aggressive calls? Do these predictions hold in the natural usage of the two types of calls? Do the predictions support the results? If not, what are some plausible explanations?
Other points:
The first sentence of the Simple Summary is confusing. I’d end the first line with a period instead of a comma and make the second line a new sentence.
The last sentence of the simple summary does not add anything, and I’d eliminate it.
ll. 74-76 this sentence does not add much unless this is related explicitly to the two different aggressive calls studied here.
l. 115 Please give the rationale for these predictions. It is not clear what they are based on.
The scatter plots in Figure 3 are in a different order than in Figure 2.
From the top down in Fig. 2 conditions D, C, B, A in Fig 3 D, B, C, A It is confusing for readers.
In Figure 4 the order of bar graphs is from the left D, B, A, C a different order again!
I would like to see more detail in how to interpret the heat maps in Figure 5. What do the different colors indicate? What is shown in each row? What are the brackets on the left indicating? How were the branches determined.
In Table 3 can you indicate the directionality of the results (e.g. A<D; B>D, etc.)?
Author Response
This is a potentially interesting paper using playback of two types of aggressive calls to captive bats to show the different calls induced changes in metabolism, food intake, body weight and gene expression. The paper is generally well written and interesting. However, there are several points that need further exploration before the paper can be full evaluated.
Thank you very much for your affirmation and valuable comments on our manuscript. Based on your questions and suggestions, we have carefully revised the manuscript.
I was confused by the initial use of “syllables” since this can have both a quantitative (how many?) and qualitative (different acoustic structure) meaning. I would drop the term “syllable” and describe the two calls as “noisy calls” and “tonal calls” throughout. This is much clearer to readers.
Thank you very much for your advice. The word “syllables” in our manuscript has a qualitative (different acoustic structure) meaning. We categorized the aggressive calls into tonal syllable aggressive calls and noisy syllable aggressive calls using syllable types of aggressive calls of V. sinensis. We have introduced the term syllables in lines 97–100 of the revised manuscript and described the difference between the two syllables of aggressive calls, and it now reads (revised portions in italics):
“Previous research has shown that tonal syllable aggressive calls (also called “tonal calls”) and noisy syllable aggressive calls (also called “noisy calls”) reflect different competitive intentions in the social conflict resolution strategies of female V. sinensis.”
In the rest of the manuscript, according to your suggestion, the phrase “tonal calls” was used to replace “tonal syllable aggressive calls”, and the phrase “noisy calls” was used to replace “noisy syllable aggressive calls”.
The authors do not help us distinguish between the contexts in which the two types of calls are given. More behavioral detail about these contexts would help interpret the results. Is one type used in mild aggression and the other in contexts leading to physical aggression? This additional information is important to include.
Revised. We have added information about on the behavioral detail about of the contexts of the two calls in V. sinensis at line in lines 100–107 of the revised manuscript, and it now reads:
“One study showed that in a crowded roost of V. sinensis, bats pushed each other; residents frequently produced aggressive calls when they were pushed by intruders over their forearm or head for a better roosting site. When the residents emitted aggressive calls with more or a higher ratio of tonal syllables, the intruders continued to push them, but when they emitted aggressive calls with more or a higher ratio of noisy syllables, the intruders stopped pushing. This suggests that the number of noisy/tonal syllables in aggressive calls in V. sinensis is a signal of intent to attack [23].”
In the methods were animals housed individually or in groups? If individually, the contexts for the different calls may not be natural. If group housed, the cage dimensions of 0.5m3 seem quite small for group housing.
Before recording the bats’ aggressive calls, the bats were kept together. Because the V. sinensis weigh no more than 30 g, they are highly clustered in the wild, and previous experience has indicated that a 0.5-m3 cage with no more than 15 individuals can guarantee the normal life of each bat. In the process of recording the aggressive calls, we used the “resident–intruder paradigm” method to record the aggressive calls: a bat was put in a cage alone as a resident, and then an "intruder" was introduced; residents made aggressive calls as the intruder pushed them. This is a classic way to record aggressive calls, and is also a way to identify who is emitting the calls.
The protocol for presenting the playbacks is not described. Were these continuous for 24h a day as the figure implies or were they played episodically? If the latter, was the playback pattern representative of the natural use of these calls? This needs to be clarified. If the playback supersaturated the normal rate of calling one would expect increased levels of stress, but these might not be natural but rather artifacts of the methods used.
We apologize for not clarifying the exact details of the playback. We have added some detailed descriptions of the playback process. We did not broadcast loud or continuous sounds, and there was a 17.5 s interval between two playback files, as now mentioned in lines 171–173 of the revised manuscript. The sentence has been corrected in the revised manuscript and now reads:
“All playback files had a syllable interval of 0.009 s, a sentence interval of 0.094 s, and an interval of 17.5 s [23]”
The playback was conducted from 8:30 to 18:30 every day. We have added information on the protocol of the playback in lines 197–199 of the revised manuscript, which now reads:
“During the playback experiment, the playback was conducted from 8:30 to 18:30 every day. Noisy calls, tonal calls, and white noise were broadcasted to groups A, B, and C, respectively.”
The playback did not cause unnatural stress. The V. sinensis we studied were a very large population, with more than 10,000 bats living under an overpass. Because the central habitat is warmer and safer, bats often compete with each other for the central habitat, and during the daytime large numbers of bats crowded together, emitting aggressive calls frequently. As far as we know, the traffic noise around the bats under the overpass is as high as 63 dB. We measured the sound intensity of the bats’ aggressive calls 1 m away from the microphone, and the average sound intensity was greater than 70 dB. Therefore, we used 70 dB as the playback sound intensity.
I’m not competent to evaluate the section on gene expression but hope that another reviewer can address this better. Many comparisons are made between different playback conditions but were corrections made for multiple statistical tests? Functional labels are given for the various genes, presumably based on function in humans, but do we know if these same genes have a similar function in the bats?
Genes with the same name that belong to mammals are likely to have similar functions. Our current understanding of genes and pathways is based on functional annotations, which are recognized by current research and may lead to new functional discoveries with further research.
In the discussion the authors are surprised by the greater response to white noise compared to call playbacks, but white noise is presumably novel for the bats, whereas the calls are familiar and have a normal context, so this might lead to greater stress response to white noise.
White noise was set up for the following reason: white noise is a non-informative call, but aggressive calls are informative calls and signal competing intentions to bats; hence, white noise was used as a control. We predicted that aggressive calls with information would have a greater impact on bats than white noise. However, surprisingly, our results did not support our prediction.
The authors make reference to Morton’s motivational-structural rules in the Introduction. Do these rules lead to different predictions about the two types of aggressive calls? Do these predictions hold in the natural usage of the two types of calls? Do the predictions support the results? If not, what are some plausible explanations?
We made these predictions based on our hypothesis that the noisy aggressive calls were more aggressive and expressed stronger social pressure than the tonal aggressive calls. We hypothesized that noisy aggressive calls are more aggressive because first, according to the motivational structure hypothesis, animals are more likely to make noisy calls when they are aggressive. Second, according to previous research, V. sinensis produce noisy aggressive calls when they are provoked, so that intruders won't push them.
We are not entirely sure whether these predictions hold in the natural usage of the two types of calls because our results did not support our hypothesis. One possible reason is that the playback time was relatively long, and in the absence of other specific stimuli, there may be habituation.
Other points:
The first sentence of the Simple Summary is confusing. I’d end the first line with a period instead of a comma and make the second line a new sentence.
Thank you for your advice; we have ended the first line with a period and made the second line a new sentence.
The last sentence of the simple summary does not add anything, and I’d eliminate it.
Thank you for your advice; we have removed the last sentence of the simple summary.
- 74-76 this sentence does not add much unless this is related explicitly to the two different aggressive calls studied here.
Thank you for your advice; we have deleted the content in lines 74–76.
- 115 Please give the rationale for these predictions. It is not clear what they are based on.
Our prediction that noisy calls have a stronger effect on food intake, body mass, etc. than tonal calls are based on our hypothesis (i.e., noisy calls have a higher aggressive intent and induce a stronger stress response in V. sinensis than tonal calls). Our hypothesis is based on the motivational structure hypothesis and previous research (Zhao et al., 2019). According to the motivational structure hypothesis, animals are more likely to make noisy calls when they are aggressive. According to previous research, V. sinensis produce noisy aggressive calls when they are provoked so that intruders won't push them. We have described the two types of aggressive calls in more detail in lines 97–109 of the revised manuscript so that the reader can better understand the hypothesis and predictions presented here.
“Previous research has shown that tonal syllable aggressive calls (also called “tonal calls”) and noisy syllable aggressive calls (also called “noisy calls”) reflect different competitive intentions in the social conflict resolution strategies of female V. sinensis [23]. One study showed that in a crowded roost of V. sinensis, bats pushed each other; residents frequently produced aggressive calls when they were pushed by intruders over their forearm or head for a better roosting site. When the residents emitted aggressive calls with more or a higher ratio of tonal syllables, the intruders continued to push them, but when they emitted aggressive calls with more or a higher ratio of noisy syllables, the intruders stopped pushing. This suggests that the number of noisy/tonal syllables in aggressive calls in V. sinensis is a signal of intent to attack [23]. Compared to tonal calls, noisy calls indicate higher aggressive intentions, and these acoustic signals may reduce physical contact and resolve conflicts in a timely manner to minimize energy consumption [23].”
The scatter plots in Figure 3 are in a different order than in Figure 2.
Revised. The revised figure is shown below.
Figure 2. Change in food intake of bats in different groups. The playback period is represented by the dark grey box. A represents the noisy calls playback group, B represents the tonal calls playback group, C represents the white noise playback group, and D represents the control group. The bar chart shows the results of the Kruskal-Wallis test, and the data are presented as mean ± standard error of the mean. * means P < 0.05, ** means P < 0.01, and *** means P < 0.001.
Figure 3. Change in body mass of bats in different groups. The playback period is represented by the dark grey box. A represents the noisy calls playback group, B represents the tonal calls playback group, C represents the white noise playback group, and D represents the control group. The bar chart shows the results of the Kruskal-Wallis test, and the data are presented as mean ± standard error of the mean. * means P <0.05, and *** means P < 0.001.
From the top down in Fig. 2 conditions D, C, B, A in Fig 3 D, B, C, A It is confusing for readers.
Revised. Please see the revised figure in the response to the previous question for details.
In Figure 4 the order of bar graphs is from the left D, B, A, C a different order again!
Revised. The revised figure is shown below.
Figure 4. Change in corticosterone concentration of bats in different groups. The playback period is represented by the dark grey box. A represents the noisy calls playback group, B represents the tonal calls playback group, C represents the white noise playback group, and D represents the control group. The bar chart shows the results of the Kruskal-Wallis test, and the data are presented as mean ± standard error of the mean.
I would like to see more detail in how to interpret the heat maps in Figure 5. What do the different colors indicate? What is shown in each row? What are the brackets on the left indicating? How were the branches determined.
In Figure 5, each column represents a sample and each row represents a gene. The colors in the figure represent the expression levels of genes in the sample: yellow represents the high expression level of the gene in the sample and blue represents the low expression level of the gene in the sample. On the left side is the tree diagram of gene clustering. The closer the distance between the branches of two genes, the closer the expression levels of the two genes. Above is the tree diagram of sample clustering. The closer the branches of two samples, the closer the expression patterns of all the genes in the two samples; that is, the closer the change in trend of the gene expression quantity.
In Table 3 can you indicate the directionality of the results (e.g. A<D; B>D, etc.)?
Table 3 is the result of the enriched pathway of the differentially expressed genes between each group, and we have sorted the number of pathways based on your recommendations to make it more friendly to the readers; the modified table is shown below.
Table 3. Kyoto Encyclopedia of Genes and Genomes pathways of differentially expressed genes between bats in different comparative groups.
|
Pathway described |
ID |
P-value |
Number of DEGs |
|
A vs D |
|
|
|
|
Folate biosynthesis |
ko00790 |
0.001094 |
2 |
|
Glycerolipid metabolism |
ko00561 |
0.006939 |
2 |
|
C vs D |
|
|
|
|
Complement and coagulation cascades |
ko04610 |
0.006103 |
2 |
|
Bile secretion |
ko04976 |
0.007234 |
2 |
|
A vs B |
|
|
|
|
Arachidonic acid metabolism |
ko00590 |
0.001225 |
2 |
|
Ferroptosis |
ko04216 |
0.001382 |
2 |
|
Mineral absorption |
ko04978 |
0.002842 |
2 |
|
B vs C |
|
|
|
|
Bile secretion |
ko04976 |
0.000972 |
4 |
|
Progesterone-mediated oocyte maturation |
ko04914 |
0.001312 |
4 |
|
Oocyte meiosis |
ko04114 |
0.002818 |
4 |
|
Mineral absorption |
ko04978 |
0.007589 |
3 |
|
Platinum drug resistance |
ko01524 |
0.009405 |
3 |
|
Phenylalanine metabolism |
ko00360 |
0.009523 |
2 |
|
A vs C |
|
|
|
|
Cocaine addiction |
ko05030 |
0.000228 |
4 |
|
Mineral absorption |
ko04978 |
0.000632 |
4 |
|
Steroid hormone biosynthesis |
ko00140 |
0.001166 |
3 |
|
Cholesterol metabolism |
ko04979 |
0.001601 |
3 |
|
Alcoholism |
ko05034 |
0.005462 |
4 |
|
Amphetamine addiction |
ko05031 |
0.008076 |
3 |
|
Phenylalanine metabolism |
ko00360 |
0.009040 |
2 |
|
Fat digestion and absorption |
ko04975 |
0.009533 |
2 |
|
B vs D |
|
|
|
|
Viral protein interaction with cytokine and cytokine receptor |
ko04061 |
0.000005 |
5 |
|
Chemokine signaling pathway |
ko04062 |
0.000051 |
6 |
|
Leishmaniasis |
ko05140 |
0.000065 |
5 |
|
Cytokine-cytokine receptor interaction |
ko04060 |
0.000085 |
6 |
|
Cell adhesion molecules |
ko04514 |
0.000421 |
5 |
|
Pathogenic Escherichia coli infection |
ko05130 |
0.000515 |
6 |
|
Th1 and Th2 cell differentiation |
ko04658 |
0.000599 |
4 |
|
NOD-like receptor signaling pathway |
ko04621 |
0.000619 |
5 |
|
Hematopoietic cell lineage |
ko04640 |
0.000671 |
4 |
|
Leukocyte transendothelial migration |
ko04670 |
0.000876 |
4 |
|
Chagas disease |
ko05142 |
0.000876 |
4 |
|
Rheumatoid arthritis |
ko05323 |
0.001321 |
4 |
|
Influenza A |
ko05164 |
0.003221 |
4 |
|
Staphylococcus aureus infection |
ko05150 |
0.006513 |
3 |
|
Th17 cell differentiation |
ko04659 |
0.007996 |
3 |
|
Caffeine metabolism |
ko00232 |
0.008818 |
1 |
|
Phagosome |
ko04145 |
0.009057 |
4 |
A vs D: the noisy calls playback group vs the control group; B vs D: the tonal calls playback group vs the control group; C vs D: the white noise playback group vs the control group; A vs B: the noisy calls playback group vs the tonal calls playback group; A vs C: the noisy calls playback group vs the white noise playback group; B vs C: the tonal calls playback group vs the white noise playback group.

Reviewer 2 Report
The study could be valuable in the ways it links animal communication, measurements of physiological stress (CORT, food intake) and gene expression. It is potentially interesting how effects can be determined via some of these metrics (gene expression) though are not apparent in others, highlighting the importance of using multiple lines of evidence in measuring stress. It is good that ‘no noise’ control was used as a baseline, in addition to the potentially stressful white noise playback.
Having said this, I have major reservations about the ethics of this study given the information provided. The authors state that they followed the ARRIVE guidelines, though these guidelines mainly concern the reporting of the study details. We are told that experimental manipulations were in accordance with the Jilin Agricultural University Animal Behavior Research Guidelines, though it is not clear whether an ethical review was conducted and approved, and whether any relevant documentation is available. I am not convinced I would obtain approval for this type of experiment at my institution (neither would I be comfortable doing it). Subjecting bats to aggressive calls and to white noise repeatedly while in cages could be considered unethical by many. The ARRIVE guidelines state that experimental details are provided so that others can replicate the experiment. Specifically, when and how often were experimental procedures conducted, and how do the experimental conditions correspond with what the bats might experience in the wild (i.e. provide rationale for the procedures)? The reader needs to know the following. What was the playback intensity compared with call intensity? How often do bats emit aggressive calls in the wild (text implies aggressive calls are rarely emitted at night, though may be frequently emitted during the day to ‘compete for habitat resources’ (how does this happen during the day? What are the ‘habitat resources’?)) At present I struggle to determine how frequently the playbacks were conducted, and how this might be representative of the natural environment, i.e. are the bats subjected to levels of stress they would not experience in nature (and no doubt this may be exacerbated by them being in small cages). What is the relevance of playing back white noise given that bats do not experience this in nature, and it is potentially stressful as a novel stimulus? These issues need to be resolved before the paper can be considered for publication.
A few additional specific comments at this stage:
The English in the Simple Summary needs improving.
Line 127 - mist nets.
Use ‘body mass’ rather than ‘weight’ throughout.
Figure 1 should also show waveforms so the reader can evaluate if the recordings were overloaded.
Figures – avoid mixing red and green, as this may be problematic for readers with limitations to their colour vision.
What are the individual data points in Fig. 2, and what do the bars and error bars represent?
Author Response
Comments and Suggestions for Authors
The study could be valuable in the ways it links animal communication, measurements of physiological stress (CORT, food intake) and gene expression. It is potentially interesting how effects can be determined via some of these metrics (gene expression) though are not apparent in others, highlighting the importance of using multiple lines of evidence in measuring stress. It is good that ‘no noise’ control was used as a baseline, in addition to the potentially stressful white noise playback.
We are glad that you think our paper is valuable and interesting. Based on your questions and suggestions, we have carefully revised the manuscript.
Having said this, I have major reservations about the ethics of this study given the information provided. The authors state that they followed the ARRIVE guidelines, though these guidelines mainly concern the reporting of the study details. We are told that experimental manipulations were in accordance with the Jilin Agricultural University Animal Behavior Research Guidelines, though it is not clear whether an ethical review was conducted and approved, and whether any relevant documentation is available. I am not convinced I would obtain approval for this type of experiment at my institution (neither would I be comfortable doing it). Subjecting bats to aggressive calls and to white noise repeatedly while in cages could be considered unethical by many.
The V. sinensis in our study are not endangered, so no special permission was required for this study. Nevertheless, our research was approved by the ethics committee of Jilin Agricultural University. We submitted the application documents for ethical approval and received approval before conducting the experiment. We have also submitted a photocopy of the ethics approval form to the system. Additionally, V. sinensis normally roost under overpasses. They are used to traffic noise (approximately 70 dB in intensity). In both wild roost and in the laboratory, our previous study confirmed that they frequently emit aggressive calls, which they themselves find loud. Thus, we thought that the playback of aggressive calls and white noise may not have any ethical issues.
The ARRIVE guidelines state that experimental details are provided so that others can replicate the experiment. Specifically, when and how often were experimental procedures conducted, and how do the experimental conditions correspond with what the bats might experience in the wild (i.e. provide rationale for the procedures)? The reader needs to know the following. What was the playback intensity compared with call intensity? How often do bats emit aggressive calls in the wild (text implies aggressive calls are rarely emitted at night, though may be frequently emitted during the day to ‘compete for habitat resources’ (how does this happen during the day? What are the ‘habitat resources’?)) At present I struggle to determine how frequently the playbacks were conducted, and how this might be representative of the natural environment, i.e. are the bats subjected to levels of stress they would not experience in nature (and no doubt this may be exacerbated by them being in small cages). What is the relevance of playing back white noise given that bats do not experience this in nature, and it is potentially stressful as a novel stimulus? These issues need to be resolved before the paper can be considered for publication.
We apologize for not clarifying the exact details of the playback procedures. We have added some detailed descriptions of the playback experiment.
We did not broadcast loud or continuous sounds, and there was a 17.5 s interval between two playback files. The sentence has been corrected in the line 171-173 of the revised manuscript and now reads:
“All playback files had a syllable interval of 0.009 s, a sentence interval of 0.094 s, and an interval of 17.5 s.”
We have added information on the protocol of the playback in lines 197–199 of the revised manuscript, which now reads:
“During the playback experiment, the playback was conducted from 8:30 to 18:30 every day. Noisy calls, tonal calls, and white noise were broadcasted to groups A, B, and C, respectively.”
The playback did not cause unnatural stress. The V. sinensis we studied were a very large population, with more than 10,000 bats living under an overpass. Because the central habitat is warmer and safer, bats often compete with each other for the central habitat. We are not sure how often an individual emitted aggressive calls, but during the daytime, many individuals huddle together and frequently produce aggressive calls. As far as we know, the traffic noise around the bats under the overpass is as high as 63 dB. We measured the sound intensity of the bats’ aggressive calls 1 m away from the microphone, and the average sound intensity was greater than 70 dB. Therefore, we used 70 dB as the playback sound intensity.
White noise was set up for the following reason: white noise is a non-informative call, but aggressive calls are informative calls and signal competing intentions to bats; hence, white noise was used as a control.
A few additional specific comments at this stage:
The English in the Simple Summary needs improving.
We have improved the English in the Simple Summary. The revised simple summary now reads (revised portions in italics):
“Most social animals have to face the social stress caused by territorial conflicts. To save costs, social animals use acoustic signals instead of physical fights to solve conflicts. Bats live in clusters and frequently produce aggressive calls of different syllables, but little is known about the effects of social stress represented by different types of aggressive calls on the physiology of bats. Here, we conducted playback experiments to investigate the effects of two types of aggressive calls representing different competitive intentions on food intake, body mass, hormone levels, and gene expression in Asian particolored bats (Vespertilio sinensis). Our results showed that different types of aggressive calls exerted different physiological effects on social animals. Interestingly, we found that more aggressive calls do not have a greater impact on bats.”
Line 127 - mist nets.
Revised. Thank you for pointing this out.
Use ‘body mass’ rather than ‘weight’ throughout.
Revised. We have replaced "weight" with "mass."
Figure 1 should also show waveforms so the reader can evaluate if the recordings were overloaded.
Thank you for your suggestion. The new picture shows the waveform. The revised Figure 1 is shown below.
Figure 1. Aggressive calls of Vespertilio sinensis; the three on the left are noisy syllable (noisy calls), and the three on the right are tonal syllable (tonal calls).
Figures – avoid mixing red and green, as this may be problematic for readers with limitations to their colour vision.
Revised. All the pictures involving red and green have been changed to other colors. The revised figures are shown below.
Figure 2. Change in food intake of bats in different groups. The playback period is represented by the dark grey box. A represents the noisy calls playback group, B represents the tonal calls playback group, C represents the white noise playback group, and D represents the control group. The bar chart shows the results of the Kruskal-Wallis test, and the data are presented as mean ± standard error of the mean. * means P < 0.05, ** means P < 0.01, and *** means P < 0.001.
Figure 3. Change in body mass of bats in different groups. The playback period is represented by the dark grey box. A represents the noisy calls playback group, B represents the tonal calls playback group, C represents the white noise playback group, and D represents the control group. The bar chart shows the results of the Kruskal-Wallis test, and the data are presented as mean ± standard error of the mean. * means P <0.05, and *** means P < 0.001.
Figure 4. Change in corticosterone concentration of bats in different groups. The playback period is represented by the dark grey box. A represents the noisy calls playback group, B represents the tonal calls playback group, C represents the white noise playback group, and D represents the control group. The bar chart shows the results of the Kruskal-Wallis test, and the data are presented as mean ± standard error of the mean.
Figure 5. Heatmaps based on differentially expressed genes from six pairwise comparisons of four groups. A: the noisy calls playback group, B: the tonal calls playback group, C: the white noise playback group, and D: the control group.
What are the individual data points in Fig. 2, and what do the bars and error bars represent?
The individual data points in the line chart in Figure 2 represent the average daily data of each group, while the individual data points in the bar chart in Figure 2 represent the actual value of each data point. The vertical bars in Figure 2 show the asterisks used to mark the differences between groups, and the error bar shows the standard error of each group of values.

Round 2
Reviewer 2 Report
The technical and scientific details provided are sufficient.